# Inflammation-Related Risk Loci in Genome-Wide Association Studies of Coronary Artery Disease

**DOI:** 10.3390/cells10020440

**Published:** 2021-02-19

**Authors:** Carina Mauersberger, Heribert Schunkert, Hendrik B. Sager

**Affiliations:** 1Klinik für Kardiologie, Deutsches Herzzentrum München, Technische Universität München, 80636 Munich, Germany; carina.mauersberger@tum.de (C.M.); schunkert@dhm.mhn.de (H.S.); 2DZHK (German Centre for Cardiovascular Research), Partner Site Munich Heart Alliance, 80636 Munich, Germany

**Keywords:** atherosclerosis, coronary artery disease, inflammation, genetics, genome-wide association studies

## Abstract

Although the importance of inflammation in atherosclerosis is now well established, the exact molecular processes linking inflammation to the development and course of the disease are not sufficiently understood. In this context, modern genetics—as applied by genome-wide association studies (GWAS)—can serve as a comprehensive and unbiased tool for the screening of potentially involved pathways. Indeed, a considerable proportion of loci discovered by GWAS is assumed to affect inflammatory processes. Despite many well-replicated association findings, however, translating genomic hits to specific molecular mechanisms remains challenging. This review provides an overview of the currently most relevant inflammation-related GWAS findings in coronary artery disease and explores their potential clinical perspectives.

## 1. Introduction

Atherosclerosis is a complex disorder that evolves over time into clinically manifesting vascular diseases such as carotid or coronary artery disease (CAD), which can lead to potentially lethal complications including stroke and myocardial infarction. Based on groundbreaking discoveries in the 20^th^ century, atherosclerosis was long assumed to be a solely lipid-driven disease further promoted by classical risk factors such as diabetes or hypertension. In addition, the idea of inflammation as a major contributor to atherosclerosis dates back several years, but it was not until 2017 that this was formally tested by the Canakinumab Antiinflammatory Thrombosis Outcome Study (CANTOS), a randomized, controlled trial (RCT) showing a beneficial effect of the anti-inflammatory drug canakinumab in CAD patients, notably without influencing lipid metabolism [1]. Soon thereafter, low dose colchicine, another drug with anti-inflammatory properties, showed a promising effect in patients after myocardial infarction and recently also in patients with chronic coronary syndrome [2,3]. Unlike canakinumab, which failed to gain approval from the U.S. and European regulatory agencies for the cardiovascular indication, colchicine is inexpensive, can be administered orally and seems to have a better chance of moving into clinical use for preventing cardiovascular events in patients at atherosclerotic risk. However, colchicine’s mechanism of action is still not completely understood. Moreover, other anti-inflammatory drugs like methotrexate failed to show beneficial outcomes in a large RCT [4].

Consequently, expanding the currently approved therapeutical options, which are mostly limited to containing traditional modifiable risk factors, is still of great importance for public health. In order to identify the key targets for potential pharmacological interventions, it is therefore crucial to elucidate the exact pathways that are involved in the development and progression of atherosclerosis in individual patients. Note that different inflammatory pathways may be involved at different stages of disease progression and thus require tailored treatment.

Genetic methods are one powerful and unbiased way to address this issue: on the one hand, they can help increase knowledge about disease development and progression to identify potentially druggable targets. On the other hand, they can also aid in selecting patients at high genetic risk, or, in light of precision medicine, recognizing which therapy is most suitable for a specific patient—as has been shown for aspirin [5], for example. Taking into account the current increased interest in inflammatory pathways in the context of CAD, this review will give an overview of relevant CAD risk loci that genome-wide association studies (GWAS) have found to be associated with inflammatory pathways.

## 2. Methodological Background

### 2.1. GWAS in a Nutshell

The idea behind large-scale genetic studies is to compare the genomes of individuals with a specific phenotype (like CAD) with those from subjects without this trait. Since the genomes of two individuals differ in many ways, GWAS compare a genome-spanning number of common variants, namely single nucleotide polymorphisms (SNPs) that account for over 80% of the genetic variation between individuals [6]. For this purpose, SNPs are genotyped using chip-based methods and allele frequencies are contrasted between affected individuals and controls for each SNP. Due to multiple testing, a risk variant is commonly considered significantly associated to the investigated trait if its p-value is below 5 × 10^−8^ [7]. While 10 years ago genomic studies were based on thousands of SNPs, now it is possible to examine millions, thereby producing a constantly growing number of significant hits.

Several achievements paved the way for the rapid breakthroughs of GWAS, which replaced small candidate inheritance studies with mostly inconsistent results in polygenic diseases such as atherosclerosis. Improved technologies, decreasing the costs and collaborations of researchers in large international consortia (e.g., the CARDIoGRAM Consortium in 2011) made it possible to genotype hundreds of thousands of individuals and thus raise statistical power. Additionally, the accessibility of databases, such as the international HapMap in 2007 [8] or the 1000 genomes project [9] made it possible to include high-quality data on imputed SNPs with no need to genotype them directly. Jointly, these developments led to the identification of a considerable number of well-replicated risk loci.

### 2.2. Challenges and Limitations of GWAS

A fundamental aspect of GWAS is that they are unbiased in their choice of markers, which opened up the possibility of linking previously unrelated genes to atherosclerosis and inflammation. Indeed, an impressive ~2/3 of the significantly associated SNPs found in the GWAS of CAD are not related to classical risk factors. Over 75% lie within non-coding regions of the genome, which makes it challenging to assign these risk variants to a certain molecular pathway [10]. As a result, the correlation of many known GWAS risk loci to candidate causal genes is based on proximity only and therefore carries a certain risk of wrong conclusions about the genes responsible for the association signal. It can be assumed, however, that many of these loci are enriched in regulatory DNA, namely *cis*-regulatory elements (CREs) that influence the transcription of different but not necessarily adjacent genes [11]. Additionally, as shown for one of the most profound CAD risk loci on chromosome 9p21, also long-distance interactions may play a role in regulating remote target gene expression [12]. These findings point to a wealth of further possibilities for the critical influence of GWAS loci on so far unrelated genes via regulatory mechanisms. 

Therefore, expression quantitative trait loci (eQTL) analyses that help elucidate associations between SNPs and gene expression at the mRNA level are now a common part of GWAS [13]. Another widely used method to strengthen the link between risk loci and their biological function is the integration of multi-tissue transcriptome data. Knowledge of the tissue or cell type in which the risk variant is relevant can further support translating GWAS results into biological context. There are now several databases with such information; STARNET, for example, is a repository of vascular and metabolic tissue data from 600 CAD patients [14]. An even more detailed approach still being developed seeks to implement single-cell genomics in GWAS [15]. Furthermore, additional insights can also be gained by specifically investigating regulatory gene networks in CAD that are regulated by both genetic and environmental factors [16]. Moving away from classical SNP array genotyping, whole-genome and especially whole-exome sequencing are another step to gain deeper insight into genomics of CAD. To date, however, these approaches have not yet been able to contribute significantly to the knowledge in this field—a situation that will certainly change soon as costs decrease and technology progresses.

Finally, what follows GWAS and their related analyses, is testing the functions of risk variants in a biological context by employing cell culture and subsequently mouse models which, despite certain limitations, provide a suitable preclinical model for this purpose [17]. While this task can now be addressed using sophisticated high-throughput genome editing techniques such as CRISPR-Cas9-based methods, however, there is still a huge effort behind implementing and subsequently evaluating these methods [18]. 

In summary, precisely assigning many GWAS hits to specific genes or biological functions is still pending, but future advanced methods should soon expand the scope of our knowledge in this field [19]. In what follows, we will focus on those inflammation-related CAD risk loci and annotated genes for which evidence of a role in atherosclerosis already exists—and which could therefore be highly significant for developing new therapeutical options or improving personalized treatment for patients in the nearer future. For an up-to-date overview of GWAS findings in CAD in general, we refer to Kessler and Schunkert 2020 [20].

## 3. Genetic Variants in CAD Linked to Inflammation

Below we discuss a selection of CAD risk loci, discovered in GWAS, that to date have been primarily assigned to inflammatory processes. An overview of these risk variants can be found in Table 1.

### 3.1. IL6R

A large Mendelian randomization study had already linked Interleukin-6 Receptor (IL6-R) signaling to CAD risk [46], when the large 2013 GWAS by the CARDIoGRAMplusC4D consortium discovered that a risk variant in the *IL6R* gene was significantly associated with CAD shortly after [21]. Mendelian randomization studies are used to investigate if a certain biomarker is causally involved in the development of a disease and are therefore another important genetic tool that can increase knowledge about a disease. Despite this strong genetic evidence for a causal involvement of *IL6R* in CAD, several preclinical studies investigating the functional role of interleukin-6 signaling in atherosclerosis yielded rather contradictory findings (reviewed in [47]). 

Nevertheless, two clinical studies assessing the effect of pharmacologically inhibiting IL6-R via the antibody tocilizumab have been conducted in patients with myocardial infarction, an advanced complication of CAD. The previous study by Kleveland and colleagues showed beneficial effects on the biomarkers C-reactive protein and troponin T [48], whereas the just-announced results from the ASSAIL-MI-trial were positive regarding the primary endpoint outcome (myocardial salvage index 69 vs. 64% in the placebo group (*p* = 0.042) [49]. 

The prospect of an already existing therapeutic option makes this target particularly attractive, especially since its pathway is also directly related to the interleukin (IL)-1β antibody canakinumab used in the previously mentioned CANTOS study [1]. IL6-R functions as the classical receptor for the cytokine IL-6, which is an important part of a highly pro-inflammatory signaling cascade that links the NLRP3 inflammasome, IL-1β, IL-18 and IL-6 with multiple downstream functions such as hematopoietic stem cell activation [50]. It remains to be seen whether further research can explain the genetic association between *IL6R* and CAD and translate it into benefits for affected patients.

### 3.2. ARHGEF26

As the transendothelial migration of leukocytes into atherosclerotic plaques is a fundamental part of atherosclerosis development and maintenance [51,52,53], several CAD risk variants are associated with this process. One of these risk variants lies within the *ARHGEF26* gene, found in a series of GWAS studies released in 2017 [25,26,27]. Interestingly, one of these studies underscores the roles of several hundred putative risk loci in a false discovery rate (FDR) of 5% in CAD GWAS [27]. Although they are truncated by the stringent GWAS significance threshold, these may hold untapped potential for better understanding CAD genetics. 

The protein encoded by *ARHGEF26* is known as Rho Guanine Nucleotide Exchange Factor 26 or SGEF. In vitro experiments have shown that it contributes to the formation of Intercellular Adhesion Molecule 1-induced cup-like docking structures in the endothelium, thus facilitating leukocyte transmigration [30,54]. This discovery could explain the in vivo finding that mice lacking the gene exhibit less atherosclerotic lesions in the aorta [55]. Klarin et al. suggested a gain-of-function phenotype for the causative mutation in humans which could account for the increased CAD susceptibility in risk allele carriers [26]. Further studies are required to better classify these findings in terms of relevance to humans, but it seems to be a promising new route for further research.

### 3.3. IL5

In 2011, the SNP rs2706399 residing in the *IL5* gene was shown to have genome-wide significant association with CAD in a large-scale GWAS conducted by the international IBC 50K CAD Consortium [31]. The encoded protein Interleukin-5 (IL-5) is mainly expressed by CD4+ T-helper cells and predominantly involved in eosinophil biology. In a bone marrow reconstitution mouse model with functional *Il5* knockout, *Il5* depletion significantly promoted atherosclerotic lesion formation [56]. A protective role for IL-5 in atherosclerosis has also been supported by a different murine experiment focused on innate lymphoid cells [57]. Though a recent study accordingly found larger atherosclerotic plaques at sites of oscillatory shear stress in *Il5* knockout mice on a proatherogenic background [58], the authors could not confirm a prospective correlation between baseline IL-5 plasma levels and future cardiovascular events in humans after more than 15 years of follow-up in a subgroup of the Malmö Diet and Cancer Study [59]. However, they could support a correlation between low IL-5 levels and the presence of carotid plaques at baseline. While this study has several weaknesses, IL-5 seems unlikely to be a relevant biomarker for CAD risk assessment. Nevertheless, it cannot be ruled out that its effect is mediated by a directed local effect at plaque lesion sites. Further research is therefore needed to clarify its potential involvement in disease development and progression.

### 3.4. SVEP1

In contrast to the GWAS findings mentioned above, a risk allele in the *SVEP1* gene has been significantly associated with CAD and myocardial infarction in an exome-wide association study [37]. Investigating SNPs exclusively within encoding genes makes this GWAS variant particularly attractive, as the process of linking the relevant SNPs to functional genes is less complex and thus less prone to misclassification. However, to date only a small number of significant hits have been identified using this approach, and *SVEP1* is the only one of these genes linked to inflammatory pathways. *SVEP1* encodes for a soluble extracellular matrix protein that binds Integrin α9β1 on lymphatic endothelial cells, thereby promoting neutrophil transendothelial migration [60]. Furthermore, *SVEP1* may also influence vascular barrier integrity via the Angiopoietin-2–Tie1/Tie2 axis [61].

In a recent study by our group, *Svep1* knockdown in a proatherogenic mouse model led to larger plaque formation and increased aortic leukocyte accumulation compared to the control group [36]. As a potentially causal factor for this phenotype, we identified elevated levels of the leukocyte attractant chemokine CXCL1 in vitro in the context of *Svep1* deficiency. However, further studies are needed to clarify the impact of this interesting finding in humans.

### 3.5. CXCL12

One of the first CAD GWAS to detect a specific variation in an inflammatory pathway locus was the GWAS published by Samani et al. in 2007 [38]. It found a likely association of variants in the *CXCL12* gene with CAD, which was later confirmed in a larger GWAS [29]. The encoded protein CXCL12, also known as Stromal Derived Factor 1 (SDF-1), is a chemokine with various molecular functions ranging from involvement in hematopoiesis to lymphocyte chemotaxis [62,63]. Its role in the context of atherosclerosis, however, has not yet been fully clarified. While the two main risk alleles have been associated with higher CXCL12 plasma levels [64] and two recent studies in mice suggest CXCL12 promotes atherosclerosis [65,66], the CXCL12 receptor CXCR4 has been shown to be atheroprotective in studies, when its function was being disturbed [67,68]. In mice with myocardial infarction, however, pharmacologically inhibiting CXCR4 has been shown to be beneficial due to improved infarct repair by augmented regulatory T cell function [69]. Therefore, further studies are needed to elucidate the exact mechanism and direction by which this locus affects CAD.

### 3.6. SH2B3/LNK

The *SH2B3*/*LNK* gene has been genome-wide significantly associated with CAD via the lead SNP rs3184504 in a GWAS carried out by the CARDIoGRAM Consortium by Schunkert et al. in 2011 [29]. The gene encodes for an intracellular negative regulator protein influencing cytokine signaling pathways. Predominantly expressed in hematopoietic cells, SH2B Adaptor Protein 3 (SH2B3) plays a particularly important role in inflammation, for example in dendritic cell activation [70], but also inhibits TNF signaling within the endothelium [71]. Interestingly, the risk locus has also been associated with not only other inflammatory diseases such as coeliac disease [72], type 1 diabetes [73] and rheumatoid arthritis [32], but also blood pressure [74,75], elevated eosinophil numbers [76] and increased platelet counts [77], which makes it more difficult to determine the relevant function of this locus with regard to CAD.

Nevertheless, similar to the effect of the human risk allele, *Sh2b3*/*Lnk* knockout in mice was associated with reduced thrombus stability and enhanced arterial thrombosis [78], but also promoted atherosclerosis in mice with hematopoietic *Sh2b3*/*Lnk* deficiency and hypercholesterolemia [79]. Additionally, in a rat myocardial infarction model *Sh2b3*/*Lnk* knockout increased fibrosis and leukocyte infiltration in the heart, which led to subsequently impaired cardiac function [80]. Thus, there is vast evidence indicating that *SH2B3*/*LNK* is crucially involved in CAD, making this gene a promising candidate for developing possible therapeutic interventions.

### 3.7. PECAM1

Another GWAS discovery affecting vascular integrity is that of the risk locus rs1867624 revealed in a study conducted by Howson et al. in 2017 [45]. It is considered to be the target of the upstream gene *PECAM1* encoding for Platelet Endothelial Cell Adhesion Molecule 1 (PECAM-1). Being expressed by endothelial and hematopoietic cells, this adhesion molecule is an important component of intercellular junctions and is critically involved in transendothelial migration of leukocytes [81]. A small prospective study found elevated PECAM-1 plasma levels in patients with acute myocardial infarction [82]. While preclinical studies in mice suggest PECAM-1 promotes atherosclerotic lesion formation at sites of disturbed flow [83], it has also demonstrated anti-inflammatory properties in other vascular regions [84]. Whether PECAM-1 is a potential biomarker or drug target for CAD remains to be clarified in further studies.

### 3.8. Further CAD Risk Loci Associated with Inflammation

While the most striking GWAS findings with relevance to inflammation have been discussed above, other risk loci with less evidence for functional involvement in CAD, to date, are worth mentioning. One such locus is located in *ITGB5*, a gene encoding Integrin Beta 5 and reported along with the lead risk locus in *ARHGEF26* in the 2017 GWAS series [25,26,27]. Integrin Beta 5 is causally involved in the phagocytosis of apoptotic cells [24], but its role in atherosclerosis has not been further explored.

Additionally, a risk locus in *MRAS* has repeatedly been genome-wide significantly associated to CAD [28,29]. The encoded protein is a member of the Ras Superfamily of Small GTPases and plays an important role in several signal transduction pathways. It has been shown to be involved in the TNF-α-stimulated activation of the integrin Lymphocyte Function-Associated Antigen 1 (LFA-1) in splenocytes [85], suggesting a role in leukocyte emigration from the spleen. Interestingly, in a canonical pathway analysis performed in the 2013 GWAS by the CARDIoGRAMplusC4D consortium [21], *MRAS* was clustered together with *IL6R* and *PLG* in the Acute Phase Response Signaling Pathway. The protein plasminogen, encoded by *PLG*, is primarily known for its role in fibrinolysis, but it has also been shown to participate in degradation of extracellular matrices and leukocyte migration [33].

Plasminogen interacts with several proteins of the complement cascade [86], which here may serve as a transition to *C1S* and *C2* presenting as additional genes associated with CAD in GWAS. They encode components of the complement system important for stimulating phagocytosis, but their functional role in atherosclerosis has not yet been investigated. The association between *C2* and CAD was described in a 2017 GWAS by Webb et al., which also investigated pleiotropy in multiple risk loci [32]. Interestingly, in this approach, the authors found *C2* to be associated with the traditional risk factor type 2 diabetes and autoimmune diseases. This work also found the *HDAC9* risk locus discovered in 2013 [21] to be associated with stroke. Furthermore, the protein encoded by *HDAC9*, Histone Deacetylase 9, was recently revealed to be involved in CAD, specifically by influencing inflammatory responses in macrophages and endothelial cells via IκB–Kinase activation [34].

Another gene relevant for endothelial function is *JCAD*, also referred to as *KIAA1462*. This gene was found to be genome-wide significantly associated with CAD in the GWAS approaches of the Coronary Artery Disease C4D Genetics Consortium and Erdmann et al. in 2011 [40,41], and subsequent experimental studies have uncovered the function of its encoded protein Junctional Cadherin 5 Associated (JCAD) [39,87]. Accordingly, *JCAD* is likely another gene harboring a risk locus that influences transendothelial migration via the activation of inflammatory processes in endothelial cells.

Finally, another very important association study for inflammation-related CAD risk loci was published in 2018 by van der Harst et. al. [23]. This study disclosed several gene variants, attributed to the *PRKCE, CFTR, PRXL2A, TRIM5, TRIM6, TRIM22, C1S* and *DHX58* genes, that had not previously been associated with CAD. Although a good base of functional data is already available for some of these candidate causal genes (for an overview of their functions, please see Table 1), experimental data that can adequately explain their function in CAD are not yet available.

### 3.9. The Drop-by-Drop Start of a Great Wave?

As is becoming clear from the discussion above, there is still a lot of research to be done. While many risk loci await translation into biological contexts in subsequent studies, there are also other important factors to be considered. CAD is a complex disease, and not only endogenous, but also environmental factors can influence the process of genotype-to-phenotype-translation—a phenomenon called gene–environment interaction (GxE). Environmental factors can promote or inhibit a disease phenotype, like the occurrence of clinical symptoms, in varied and time-dependent ways, which seems to be particularly distinct in risk loci affecting later phases of disease progression [88]. Put more pointedly, GxE could lead to a situation in which individuals with identical GWAS SNP patterns exposed to different environmental influences are enrolled in both the case and the control groups of a study. Consequently, such affected risk variants are unlikely to meet the required significance threshold and thus may be under-represented in GWAS. Scientists address this issue by integrating best-possible environmental, but also gene expression data in their GWAS [89].

To conclude, it is evident that GWAS alone cannot entirely decipher CAD. Research therefore continues to move from classical GWAS models to the realization of a more comprehensive concept: systems genetics. This powerful approach aims to cohesively combine statistical and experimental data such as genomics, epigenomics, proteomics and metabolomics [90]. As mentioned above, a common approach linking GWAS and transcriptomics is eQTL analysis, integrating gene expression data based on mRNA levels. In contrast, incorporating protein expression into multi-omics studies in so-called pQTL approaches is less developed thus far. Linking such information to genetic findings can contribute to associating risk loci to candidate causal genes [91] or help identify related druggable targets [92]. Further implementation of these comprehensive approaches, which are currently still mostly in their infancy, is therefore eagerly awaited in future studies. Such advances will likely bring to light a multitude of important answers to our current questions, particularly with regard to inflammation-related variants. 

## 4. Clinical Applications of GWAS Findings

One hope that has driven GWAS from their inception is the ability to identify previously unknown, critical disease pathways that can enable more comprehensive pharmacological targeting. The above discussion indicates that this is still a long way off. But it is not the only possible application of GWAS findings (Figure 1): Meanwhile, other promising approaches to the clinical use of GWAS results have also taken the stage, as will be discussed in the following.

### 4.1. Polygenic Risk Scores

A criticism of GWAS is that they almost exclusively detect common mutations with rather moderate effects on disease. This could change as more individuals are tested and less frequent variants therefore reach the required significance threshold. But while rare mutations might be responsible for very severe disease outcomes, they only affect a minority of patients. CAD, however, affects innumerable people worldwide and is therefore promoted by common mutations which, considered separately, may indeed have only small effects on the CAD risk. Yet, in their large quantities and critical combinations—it is estimated that the average European carries hundreds of risk alleles in the genome—these common mutations appear to be responsible for a substantial amount of CAD heritability. Therefore, one very interesting approach that takes into account the multitude of genetic influences on CAD is polygenic risk score (PRS) assessment. The first truly helpful PRS approach was published in 2018 by Khera et al. and impressively included more than six million variants weighted by their number and impact on CAD risk in an individual [93]. Their PRS identified a three-fold increased CAD risk for 8% of the population, which is twenty times more than the carrier frequency of familial hypercholesterolemia, for instance. Broad clinical application of such scores could help identify individuals with very high genetic risk at an early age, before the disease becomes symptomatic, and thus counteract the disease course through appropriate lifestyle adjustments or pharmacological treatment. For example, physical activity or early onset statin therapy can remarkably reduce risk in highly affected individuals [94].

Applying PRS raises potential consequences for each given risk value that must be well considered. For one, we need appropriate therapeutic approaches, and this requires more than reducing classical risk factors, for which a correlation of only about 1/3 of the genetic risk has been proven. Furthermore, this knowledge must not develop into enforced therapeutic monitoring of or discrimination against those affected from birth. 

Another intriguing method described in a recently published study used a coronary disease genetic risk score to investigate the genetic relationship between CAD and other diseases [95]. In addition to identifying an association linking CAD to classical risk factors and diseases secondary to CAD, the authors found that their CAD PRS was associated with a reduced risk of migraine. Polygenetic risk scores could therefore also help clarify disease interrelationships and more comprehensively understand CAD.

### 4.2. Precision Medicine

Knowledge about a particular risk factor, especially when appropriately applied to some of the risk alleles mentioned above, may help tailor therapy and could also explain contradictory findings in different studies. Therefore, in future clinical trials, risk factor assessment could aid in better weighing the risk–benefit ratio of drug treatment for each individual and also facilitate the approval of therapies that have not yet been proven to be beneficial within a certain indication. Several post hoc analyses of clinical trials have already shown this to be a valuable approach worth pursuing further. For example, as mentioned before, carriers of the risk allele in the *GUCY1A1* gene attain greater benefit from aspirin treatment [5], and people with high CAD polygenic risk scores profit more from treatment with statins [96,97] and PCSK9 inhibitors [98,99]. Notably, also the CANTOS trial specifically investigated the influence of canakinumab on patients with increased high-sensitivity C-reactive protein (hsCRP) levels [1]. In this context, more exploration of inflammation-related GWAS hits for CAD may not only shed light on relevant genetic risk factors but potentially also facilitate the identification of previously unknown—and possibly more specific—biomarkers to optimize treatment for individual patients.

## 5. Conclusions

GWAS identified large numbers of risk variants in genes with previously unknown contributions to CAD, thereby increasing our knowledge of the disease and steering research in new directions. Moreover, they also aid in finding new drug targets for preventing or treating the disease. In fact, most GWAS risk loci did not correlate with previously studied targets and helped increase awareness of how other pathways, such as inflammation, contribute to CAD. However, the majority of GWAS findings involve non-coding regions of the genome, which makes connecting these variants to a specific gene or biological function quite complicated. Therefore, the greatest challenge now rather lies in allocating the results into biological contexts. Several approaches, such as systems genetics [89], which seek to integrate genetic findings with other functional data such as proteomics, will hopefully contribute to a better understanding of previously unclassified CAD GWAS loci, including in the context of inflammation.

Second, GWAS findings can contribute enormously to assessing individual risk for CAD even before symptom onset. GWAS research points towards the causal involvement of several risk variants, which, when considered separately, may have rather small biological effects but taken together are estimated to be responsible for about 40% of CAD heritability [10]. This is significantly more than the estimated proportion attributable to rare mutations [27]. Together with a more comprehensive therapeutic toolkit, GWAS could lead directly to identifying individuals at high genetic risk for developing CAD and help prevent adverse disease progression. Moreover, even single known mutations can contribute to a better individual therapy in the sense of precision medicine, and further add to beneficial outcomes for affected patients.

There is a great deal of research ahead to fulfill the promise of GWAS, particularly with regard to clinical applications. Hopefully, technical improvements and other ingenious future approaches will soon help improve the medical treatment of coronary artery disease, one of the biggest health problems, with one of the highest burdens of morbidity and mortality worldwide today.

## Figures and Tables

**Figure 1 cells-10-00440-f001:**
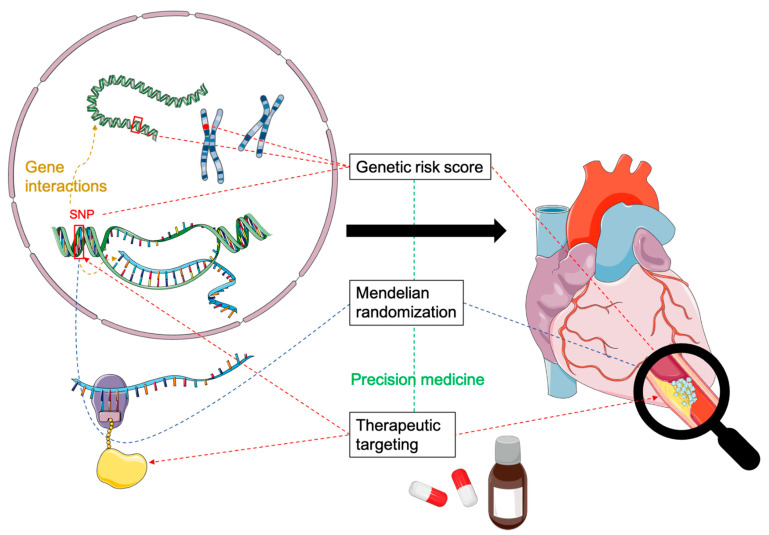
Simplified representation of the potential influence results from genome-wide significant association studies (GWAS) may have on clinical perspectives in coronary artery disease. For details, please refer to the text. Contains modified images from Servier Medical Art under a Creative Commons Attribution 3.0 Unported License.

**Table 1 cells-10-00440-t001:** Genome- and exome-wide significant risk loci associated with coronary artery disease in genes primarily related to inflammatory processes (modified from Erdmann et al. (2018) [10]. Loci that have received more extensive experimental research regarding their functional involvement in coronary artery disease (CAD) are marked in bold. GWAS: Genome-Wide Association Study, IKK: IκB Kinase, IL: Interleukin, TLR-4: Toll-Like-Receptor 4, SNP: single nucleotide polymorphism.

Locus	Lead SNP	Inflammation-Related Gene	Gene Function(s)	GWAS References
1q21.3	rs4845625	***IL6R***	IL-6 signaling, pro-inflammatory immune responses	[21]
2p21	rs582384	*PRKCE*	Cardiac muscle contraction, cardioprotection, macrophage and dendritic cell activation (TLR-4 signaling) [22]	[23]
3q21.2	rs142695226	*ITGB5*	αvβ5 integrin component, apoptotic cell phagocytosis [24]	[25,26,27]
3q22.3	rs2306374	*MRAS*	Signal transduction in cell growth and differentiation, oncogene	[28,29]
3q25.2	rs12493885	***ARHGEF26***	Macropinocytosis, leukocyte transendothelial migration [30]	[25,26,27]
5q31.1	rs2706399	***IL5***	Eosinophil biology	[31]
6p21.33	rs3130683	*C2*	Stimulation of phagocytes (complement component)	[32]
6q26	rs4252120	*PLG*	Fibrinolysis, degradation of extracellular matrices, leukocyte migration [33]	[21]
7p21.1	rs2023938	*HDAC9*	Histone deacetylase, IKK activation, inflammatory responses in macrophages and endothelial cells [34]	[21]
7q31.2	rs975722	*CFTR*	Cystic fibrosis-related chloride channel, respiratory and intestinal immune responses [35]	[23]
9q31.3	rs111245230	***SVEP1***	Leukocyte transendothelial migration [36]	[37]
10q11.21	rs1746048	***CXCL12***	Lymphocyte chemotaxis, angiogenesis	[29,38]
10p11.23	rs2505083	*JCAD/KIAA1462*	Leukocyte endothelium adhesion [39]	[40,41]
10q23.1	rs17680741	*PRXL2A/FAM213A*	Myelopoiesis, negative regulation of p53 tumor suppressor gene [42]	[23]
11p15.4	rs11601507	*TRIM5, TRIM22, TRIM6*	Innate immune responses, viral infections [43]	[23]
12p13.31	rs11838267	*C1S*	Stimulation of phagocytes (complement component)	[23]
12q24.12	rs3184504	***SH2B3/LNK***	Cytokine signaling	[29]
17q21.2	rs2074158	*DHX58*	Antiviral signaling, suppression of tumor cell growth [44]	[23]
17q23.3	rs1867624	***PECAM1***	Endothelium intercellular junctions	[45]

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
