# Peer review of "Inflammation-Related Risk Loci in Genome-Wide Association Studies of Coronary Artery Disease"

_cells, 2021, doi:10.3390/cells10020440_

Round 1

Reviewer 1 Report

I was honored to review the manuscript entitled “GWAS loci for coronary artery disease and inflammation” submitted to Cells.

I recommend to accept the manuscript after minor revision.

There are only some points to correct:

 - please provide the list of abbreviations

- discussion section needs improvement – please provide information on how your results will translate into clinical practice

- in discussion section please provide study strong points  and study limitation section

- please correct typos

 - It would be also useful to illustrate some of the explained mechanisms

I recommend to accept the manuscript after minor revision.

Author Response

Reviewer #1:

I was honored to review the manuscript entitled “GWAS loci for coronary artery disease and inflammation” submitted to Cells.

I recommend to accept the manuscript after minor revision. There are only some points to correct:

We thank this reviewer for acknowledging our work and appreciate her/his helpful comments.

Please provide the list of abbreviations

We added a list of abbreviations at the end of the manuscript (lines 440-482).

Discussion section needs improvement – please provide information on how your results will translate into clinical practice

We thank the reviewer for addressing this important point. As direct influence of inflammation-related CAD GWAS results on the development of pharmacological treatment has not been reported yet, in our manuscript, we focus on possible outlooks, and more specifically, polygenic risk score assessment and precision medicine with expected implementation in clinical practice in the nearer future. However, to highlight the relevance of clinical perspectives of GWAS, we renamed this section to “Clinical Applications of GWAS Findings” and better explained this important aspect (lines 324-333):

“Further implementation of these comprehensive approaches, which are currently still mostly in their infancy, is therefore eagerly awaited in future studies. Such advances will likely bring to light a multitude of important answers to our current questions, particularly with regard to inflammation-related variants. Meanwhile, other promising approaches to the clinical use of GWAS results have also taken the stage, as will be discussed in the following section.

  1. Clinical Applications of GWAS Findings

One hope that has driven GWAS from their inception is the ability to identify previously unknown, critical disease pathways that can enable more comprehensive pharmacological targeting. The above discussion indicates that this is still a long way off…”

Followed by the sections 4.1 Polygenic Risk Scores and 4.2 Precision Medicine.

In discussion section please provide study strong points and study limitation section

Our manuscript provides a review of inflammation related risk loci in current GWAS studies of coronary artery disease. Therefore, our first goal was to explain the general methodological background of GWAS studies, but also their limitations early in the first section following the introduction. However, to meet the reviewer’s demand, we added another section at the end of section 3 discussing further important aspects to be considered in GWAS. It reads as follows (lines 300-329):

“3.9. The drop-by-drop start of a great wave?

As is becoming clear from the discussion above, there is still a lot of research to be done. While many risk loci await translation into biological contexts in subsequent studies, there are also other important factors to be considered. CAD is a complex disease, and not only endogenous, but also environmental factors can influence the process of genotype-to-phenotype-translation – a phenomenon called gene-environment interaction (GxE). Environmental factors can promote or inhibit a disease phenotype, like the occurrence of clinical symptoms, in varied and time-dependent ways, which seems to be particularly distinct in risk loci affecting later phases of disease progression [88]. Put more pointedly, GxE could lead to a situation in which individuals with identical GWAS-SNP patterns exposed to different environmental influences are enrolled in both the case and the control groups of a study. Consequently, such affected risk variants are unlikely to meet the required significance threshold and thus may be underrepresented in GWAS. Scientists address this issue by integrating best-possible environmental and gene expression data in their GWAS [89].

To conclude, it is evident that GWAS alone cannot entirely decipher CAD. Research therefore continues to move from classical GWAS models to the realization of a more comprehensive concept: systems genetics. This powerful approach aims to cohesively combine statistical and experimental data such as genomics, epigenomics, proteomics and metabolomics [90]. As mentioned above, a common approach linking GWAS and transcriptomics is eQTL analysis, integrating gene expression data based on mRNA levels. In contrast, incorporating protein expression into multi-omics studies in so-called pQTL approaches is less developed thus far. Linking such information to genetic findings can contribute to associating risk loci to candidate causal genes in bioinformatics analyses [91] and help identify related druggable targets [92]. Further implementation of these com-prehensive approaches, which are currently still mostly in their infancy, is therefore eagerly awaited in future studies. Such advances will likely bring to light a multitude of important answers to our current questions, particularly with regard to inflammation-related variants. Meanwhile, other promising approaches to the clinical use of GWAS results have also taken the stage, as will be discussed in the following section.”

Please correct typos

The manuscript has now been proofread by a native speaker and thoroughly corrected for typos.

It would be also useful to illustrate some of the explained mechanisms

Although there is evidence from genetic studies of functional involvement of the aforementioned risk loci in CAD, the exact mechanisms of how they influence disease risk are largely unknown. Hence, we would be hesitant to provide a speculative figure illustrating the putative mechanisms.

I recommend to accept the manuscript after minor revision.

We thank this reviewer for her/his valuable comments and hope to have adequately addressed her/his suggestions in the revised version of the manuscript.

Reviewer 2 Report

This review paper is to provide an overview of findings of GWAS loci for coronary artery disease and inflammation. Unfortunately, this manuscript was not well written. The overall quality of paper was not very high. This paper does not significantly advance the knowledge in this field.

The followings are a few examples:
Please specify what anti-inflammatory drug was used in CAD patients in the randomized controlled trial (line 31, page 1).

Why canakinumab was chosen for comparison with colchicine (line 35, page 1)?

What does this statement "The SNP rs2706399 residing in the encoding IL5 gene has been significantly associated with CAD in 2011" mean (lines 115 – 116, page 3)?

What is the rational for choosing 7 SNPs out of 18 listed SNPs for detailed discussion? The section of "Genetic Variants of CAD Linked to Inflammation" should be re-organized.

Author Response

Reviewer #2:

This review paper is to provide an overview of findings of GWAS loci for coronary artery disease and inflammation. Unfortunately, this manuscript was not well written. The overall quality of paper was not very high. This paper does not significantly advance the knowledge in this field.

We thank the reviewer for her/his evaluation of the manuscript and regret that she/he is not satisfied with the writing and content of the manuscript. We hope that after revising the review we will sufficiently address the points of criticism.

The followings are a few examples:

Please specify what anti-inflammatory drug was used in CAD patients in the randomized controlled trial (line 31, page 1).

We added the name of the anti-inflammatory drug used in CANTOS in this section. It now reads (lines 29-32):

“…this was finally proven by the Canakinumab Antiinflammatory Thrombosis Outcome Study (CANTOS), a randomized, controlled trial (RCT) showing a beneficial effect of the anti-inflammatory drug canakinumab in CAD patients, notably without influencing lipid metabolism [1].”

Why canakinumab was chosen for comparison with colchicine (line 35, page 1)?

Despite the groundbreaking character of the 2017 CANTOS study for the inflammation hypothesis, Novartis, the manufacturer of canakinumab, opted not to pursue its approval for CAD after receiving a rejection letter from the U.S. Food and Drug Administration (FDA). Novartis also withdrew its application for a cardiovascular indication of canakinumab in Europe in 2018. Therefore, we have reason to conclude that a widespread use of this rather expensive drug beyond official approval by state organs for this indication is, unfortunately, very unlikely. This evaluation is shared broadly, for similar comparisons see:

  1. Sehested, T.S.G.; Bjerre, J., et al. Cost-effectiveness of Canakinumab for Prevention of Recurrent Cardiovascular Events. JAMA Cardiol 2019, 4, 128-135.
  2. Roubille, F.; Tardif, J.C. Colchicine for Secondary Cardiovascular Prevention in Coronary Disease. Circulation 2020, 142, 1901-1904.
  3. Roman, Y.M.; Hernandez, A.V., et al. The Role of Suppressing Inflammation in the Treatment of Atherosclerotic Cardiovascular Disease. Ann. Pharmacother. 2020, 54, 1021-1029.

In contrast, the alkaloid drug colchicine is orally administrable, not restricted to the patent protection of a single manufacturer, inexpensive and (like canakinumab) has shown a beneficial effect in a large randomized, controlled trial in atherosclerotic patients, but additionally also for patients within the first 30 days after myocardial infarction:

  1. Nidorf, S.M.; Fiolet, A.T.L., et al. Colchicine in Patients with Chronic Coronary Disease. N. Engl. J. Med. 2020, 383, 1838-1847.
  2. Tardif, J.C.; Kouz, S., et al. Efficacy and Safety of Low-Dose Colchicine after Myocardial Infarction. N. Engl. J. Med. 2019, 381, 2497-2505.

In our opinion, these are several reasons to believe that colchicine is more likely to be used in atherosclerotic risk patients than canakinumab. However, to meet the critique of the reviewer, we have adjusted the wording in the manuscript to better justify our basis for this comparison and it now reads (lines 35-39):

“Unlike canakinumab, which failed to gain approval from the U.S. and European regulatory agencies for the cardiovascular indication, colchicine is inexpensive, can be administered orally and seems to have a better chance of moving into clinical use for preventing cardiovascular events in patients with atherosclerotic risk. However, colchicine’s mechanism of action is still not completely understood…”

What does this statement "The SNP rs2706399 residing in the encoding IL5 gene has been significantly associated with CAD in 2011" mean (lines 115 – 116, page 3)?

This statement is intended to introduce the genomic discovery of the genome-wide significantly associated lead SNP "rs2706399" which lies within the IL5 gene and was therefore assigned to IL5. However, we assume that the reviewer is specifically targeting the last part of the corresponding sentence with this question and therefore also interferes with the request of reviewer #3 for a better explanation of the underlying GWAS. We are happy to respond to this request by giving more information to the source GWAS at the beginning of the discussion of each candidate gene in section 3, highlighting important characteristics and background information of these studies where relevant for the review (lines 131-299).

For instance, the mentioned sentence (lines 175-178) now reads:

“In 2011, the SNP rs2706399 residing in the IL5 gene was shown to have genome-wide significant association with CAD in a large-scale GWAS conducted by the international IBC 50K CAD Consortium [31]. The encoded protein Interleukin-5 (IL-5) is mainly expressed by CD4+ T-helper cells and predominantly involved in eosinophil biology…”

What is the rational for choosing 7 SNPs out of 18 listed SNPs for detailed discussion?

We thank the reviewer for pointing out this important aspect. We chose to highlight seven of the risk loci listed in table 1 because there is a base of studies providing insight into how these specific risk loci may functionally influence CAD. However, as noted in the text, many of the risk loci found in GWAS have not been adequately studied, and this is true for the missing 11 risk loci listed in table 1. Therefore, a separate speculative discussion of the affected risk alleles was not the primary goal of our review due to the lack of functional data. However, to clarify the reviewer's valid objection also in the manuscript, we briefly commented on these loci in an additional section 3.8. (lines 256-299), where we focus on the GWAS studies behind these discoveries. It reads as follows:

“3.8. Further CAD risk loci associated with inflammation

While the most striking GWAS findings with relevance to inflammation have been discussed above, other risk loci with less evidence for functional involvement in CAD, to date, are worth mentioning. One such locus is located in ITGB5, a gene encoding Integrin Beta 5 and reported along with the lead risk locus in ARHGEF26 in the 2017 GWAS series [25-27]. Integrin Beta 5 is causally involved in phagocytosis of apoptotic cells [24], but its role in atherosclerosis has not been further explored.

Additionally, a risk locus in MRAS has repeatedly been genome-wide significantly associated to CAD [28,29]. The encoded protein is a member of the Ras Superfamily of Small GTPases and plays an important role in several signal transduction pathways. It has been shown to be involved in TNF-α-stimulated activation of the integrin Lymphocyte Function-Associated Antigen 1 (LFA-1) in splenocytes [85], suggesting a role in leukocyte emigration from the spleen. Interestingly, in a canonical pathway analysis performed in the 2013 GWAS by the CARDIoGRAMplusC4D consortium [21], MRAS was clustered together with IL6R and PLG in the Acute Phase Response Signaling Pathway. The protein plasminogen, encoded by PLG, is primarily known for its role in fibrinolysis, but it has also been shown to participate in degradation of extracellular matrices and leukocyte migration [33].

Plasminogen interacts with several proteins of the complement cascade [86], which here may serve as a transition to C1S and C2 presenting as additional genes associated with CAD in GWAS. They encode components of the complement system important for stimulating phagocytosis, but their functional role in atherosclerosis has not yet been investigated. The association between C2 and CAD was described in a 2017 GWAS by Webb et al. that also investigated pleiotropy in multiple risk loci [32]. Interestingly, in this approach, the authors found C2 to be associated with the traditional risk factor type 2 diabetes and autoimmune diseases. This work also found the HDAC9 risk locus dis-covered in 2013 [21] to associate with stroke. Further, the protein encoded by HDAC9, Histone Deacetylase 9, was recently revealed to be involved in CAD, specifically by influencing inflammatory responses in macrophages and endothelial cells via IκB-Kinase activation [34].

Another gene relevant for endothelial function is JCAD, also referred to as KIAA1462. This gene was found to be genome-wide significantly associated with CAD in the GWAS approaches of the Coronary Artery Disease C4D Genetics Consortium and Erdmann et al. in 2011 [40,41], and subsequent experimental studies have uncovered the function of its encoded protein Junctional Cadherin 5 Associated (JCAD) [39,87]. Accordingly, JCAD is likely another gene harboring a risk locus that influences transendothelial migration via activation of inflammatory processes in endothelial cells.

Finally, another very important association study for inflammation-related CAD risk loci was published in 2018 by van der Harst et. al. [23]. This study disclosed several gene variants, attributed to the PRKCE, CFTR, PRXL2A, TRIM5, TRIM6, TRIM22, C1S and DHX58 genes, that had not previously associated with CAD. Although a good base of functional data is already available for some of these candidate causal genes (for an overview of their functions, please see Table 1), experimental data that can adequately explain their function in CAD are not yet available.”

Moreover, we added information regarding the association of the gene JCAD with CAD in Table 1 and our manuscript, as we regard it worth being mentioned in this context, too.

The section of "Genetic Variants of CAD Linked to Inflammation" should be re-organized.

The risk loci listed in Table 1 are sorted by their localization in the genome. In the main text, however, we chose to order the risk loci by date of first publication. To facilitate finding the explanation in the text for the reader, we have correspondingly re-organized the section according to the order in Table 1. Nevertheless, as stated in the previous explanation, the aforementioned risk loci in the new section 3.8 cannot follow this sorting, as they will be considered at the end of this subitem.

Reviewer 3 Report

Review article

The article entitled GWAS Loci for Coronary Artery Disease and Inflammation describes the use of GWAS as a source to determine correlations between genes and CAD risk.

The article is well written and contains interesting information about inflammatory genes and SNPs. The title could demonstrate that the article is a resume of different GWAS studies related to CAD (that could be added as a subtitle)

The article is a review describing the association of SNPs and inflammatory target genes and their correlation to the development of cardiovascular diseases. The authors selected some important genes correlated to CAD from the literature studied in different GWAS studies. Even though the article is an interesting resume of the recent finds it lacks to describe the importance of these discoveries, and their impact to the treatment of these diseases. A better explanation of the original article used as a source would improve the reading and reinforce the importance of the findings. The authors used the sentence “Therefore, one could argue that the biggest challenge now rather lies in the allocation of the results into a biological context.” And should try to suggest some literature were how this could be done. To describe the force of findings and correlations.

In conclusion, to improve the article findings more explanation of the GWAS sources and the correlation of these findings to practical studies should be added as a literature source.

Author Response

Reviewer #3:

The article entitled GWAS Loci for Coronary Artery Disease and Inflammation describes the use of GWAS as a source to determine correlations between genes and CAD risk.

The article is well written and contains interesting information about inflammatory genes and SNPs.

We thank this reviewer for her/his valuable comments and are pleased to respond.

The title could demonstrate that the article is a resume of different GWAS studies related to CAD (that could be added as a subtitle)

We agree with the reviewer’s opinion and changed the title of the review for an immediate understanding of its intention. It now reads: “Inflammation-Related Risk Loci in Genome-Wide Association Studies of Coronary Artery Disease”.

The article is a review describing the association of SNPs and inflammatory target genes and their correlation to the development of cardiovascular diseases. The authors selected some important genes correlated to CAD from the literature studied in different GWAS studies. Even though the article is an interesting resume of the recent finds it lacks to describe the importance of these discoveries, and their impact to the treatment of these diseases. A better explanation of the original article used as a source would improve the reading and reinforce the importance of the findings.

We are happy to respond to this request by giving more information to the source GWAS at the beginning of the discussion of each candidate gene in section 3, highlighting important characteristics and background information of these studies where relevant for the review (lines 131-299).

The authors used the sentence “Therefore, one could argue that the biggest challenge now rather lies in the allocation of the results into a biological context.” And should try to suggest some literature were how this could be done. To describe the force of findings and correlations.

We thank the reviewer for highlighting this important point. We have explored possible approaches to translating genetic discoveries into biological context in more detail in paragraph 2.2 of the manuscript. It reads as follows (lines 92-114):

“…These findings point to a wealth of further possibilities for a critical influence of GWAS loci on so far unrelated genes via regulatory mechanisms.

Therefore, expression quantitative trait loci (eQTL) analyses that help elucidate as-sociations between SNPs and gene expression at the mRNA level are now a common part of GWAS [13]. Another widely used method to strengthen the link between risk loci and their biological function is the integration of multi-tissue transcriptome data. Knowledge of the tissue or cell type in which the risk variant is relevant can further support translating GWAS results into biological context. There are now several databases with such information; STARNET, for example, is a repository of vascular and metabolic tissue data from 600 CAD patients [14]. An even more detailed approach still being developed seeks to implement single-cell genomics in GWAS [15]. Furthermore, additional insights can also be gained by specifically investigating regulatory gene networks in CAD that are regulated both by genetic and environmental factors [16]. Moving away from classical SNP array genotyping, whole-genome and especially whole-exome sequencing are another step to gain deeper insight into genomics of CAD. So far, however, these approaches have not yet been able to contribute significantly to the knowledge in this field - a situation that will certainly change soon as costs decrease and technology progresses.

Finally, what follows GWAS and their related analyses, is testing the functions of risk variants in a biological context by employing cell culture and subsequently mouse models which, despite certain limitations, provide a suitable preclinical model for this purpose [17]. While this task can now be addressed using sophisticated high-throughput genome editing techniques such as CRISPR-Cas9-based methods, however, there is still a huge effort behind implementing and subsequently evaluating these methods [18].”

In conclusion, to improve the article findings more explanation of the GWAS sources and the correlation of these findings to practical studies should be added as a literature source.

We thank this reviewer for her/his helpful remarks and hope to have adequately met her/his demands in the revised manuscript.

Round 2

Reviewer 2 Report

This revised review paper is significantly improved. I do not have further comments.

Reviewer 3 Report

The authors responded adequately to all the questions made previously, and now the article seems appropriate to be published in the present version.